# A Comparative Study on the Adipogenic Differentiation of Mesenchymal Stem/Stromal Cells in 2D and 3D Culture

**DOI:** 10.3390/cells11081313

**Published:** 2022-04-13

**Authors:** Anne Wolff, Marcus Frank, Susanne Staehlke, Kirsten Peters

**Affiliations:** 1Department of Cell Biology, Rostock University Medical Centre, 18057 Rostock, Germany; anne.wolff@med.uni-rostock.de (A.W.); susanne.staehlke@med.uni-rostock.de (S.S.); 2Medical Biology and Electron Microscopy Centre, Rostock University Medical Centre, 18057 Rostock, Germany; marcus.frank@med.uni-rostock.de; 3Department of Life, Light & Matter, University of Rostock, 18051 Rostock, Germany

**Keywords:** mesenchymal stem/stromal cells, 2D culture, 3D culture, spheroids, adipogenic differentiation, adipokines

## Abstract

Mesenchymal stem/stromal cells (MSC) are capable of renewing the progenitor cell fraction or differentiating in a tissue-specific manner. Adipogenic differentiation of adipose-tissue-derived MSC (adMSC) is important in various pathological processes. Adipocytes and their progenitors are metabolically active and secrete molecules (adipokines) that have both pro- and anti-inflammatory properties. Cell culturing in 2D is commonly used to study cellular responses, but the 2D environment does not reflect the structural situation for most cell types. Therefore, 3D culture systems have been developed to create an environment considered more physiological. Since knowledge about the effects of 3D cultivation on adipogenic differentiation is limited, we investigated its effects on adipogenic differentiation and adipokine release of adMSC (up to 28 days) and compared these with the effects in 2D. We demonstrated that cultivation conditions are crucial for cell behavior: in both 2D and 3D culture, adipogenic differentiation occurred only after specific stimulation. While the size and structure of adipogenically stimulated 3D spheroids remained stable during the experiment, the unstimulated spheroids showed signs of disintegration. Adipokine release was dependent on culture dimensionality; we found upregulated adiponectin and downregulated pro-inflammatory factors. Our findings are relevant for cell therapeutic applications of adMSC in complex, three-dimensionally arranged tissues.

## 1. Introduction

The common practice of culturing cells on a two-dimensional (2D) surface does not reflect the physiological situation of most cell types in tissues. Because cells in the body are arranged in three dimensions, a 2D cell culture surface disregards aspects such as cell polarity, the three-dimensional (3D) organization of intercellular contacts, and the extracellular milieu. Due to these limitations, 3D culture systems have been developed to provide an environment that is considered to be more physiological in terms of the structural arrangement of the cells in a body [1,2,3,4]. Several 3D cultivation techniques are known, including the use of natural and synthetic scaffolds [5,6,7], scaffold-free models (e.g., hanging drop), and hybrid model systems that use a scaffold to support scaffold-free systems [8,9].

Several recent reports have pointed out the specific capacities of 3D spheroid cultures in the differentiation of various cell types including osteoblasts [10], organotypic skin cells [11], and cancer (associated) cells [12,13,14], in addition to tailored derivatives from induced pluripotent stem cells (iPSC) [15,16] or mesenchymal stem/stromal cells (MSC) [17,18]. Three-dimensional spheroid cultures allow detailed comparison with the respective 2D culture methods, which opens further possibilities for co-culturing and putative therapeutic applications [19]. Therefore, we decided to examine adipogenic differentiation of adipose-tissue-derived mesenchymal stem/stromal cells (adMSC) in a newly adapted 3D spheroid system using cell-bound magnetizable beads to achieve a higher tissue-typical structural organization and compare the cellular responses with those obtained from the parallel 2D cultivation.

Adipogenic differentiation is characterized by changes in the sequential expression of genes that determine the adipocytic phenotype of the cells, such as peroxisome proliferator-activated receptor gamma (PPARγ). This is reflected in the appearance of various adipogenesis-specific mRNA and proteins and in the end usually leads to lipid accumulation [20]. Adipogenic differentiation not only plays a role during embryonic development and the related formation of adipose tissue, but is also important in various pathological processes such as diabetes [21] and osteoporosis [22].

Adipocytes are metabolically active and secrete pro-inflammatory cytokines, hormones, and growth factors, among other things [23]. These adipocyte-released factors are collectively known as adipokines [24]. Adipokines include adiponectin, a protein hormone involved in the regulation of glucose levels and fatty acid turnover [25]; the plasminogen activator inhibitor-1 (PAI-1), a serine protease that inhibits fibrinolysis and induces insulin resistance and metabolic deviations [26]; and the pro-inflammatory factors interleukin-6 (IL-6) and the monocyte chemoattractant protein 1 (MCP-1). Adipokines possess both pro- and anti-inflammatory properties and link energy metabolism to immunological responses [27].

adMSC are cells of the adipose tissue that can renew the mesenchymal progenitor cell fraction or may alternatively differentiate in a tissue-specific manner [28,29]. Thus, their presence is a prerequisite for the physiological turnover and regeneration of adipose tissue [30]. Like all mesenchymal stromal cells, adMSC can exhibit immunosuppressive, anti-inflammatory, and pro-angiogenic features [31,32]. Thus, they represent a promising cell type for use in regenerative therapies [33]. Because adMSC retain their ability to differentiate in vitro, they can be used to examine both essential signal transduction and the effect of pharmacological compounds.

Only a few studies have addressed the effects of 3D cultivation on the degree of differentiation of MSC compared with 2D cultivation so far. For example, Bogdanova-Jatniece et al. [34] detected the expression of the pluripotency marker Nanog in spontaneously formed 3D aggregates of adMSC, whereas 2D cultures were negative for this marker. Regarding adipogenic differentiation, there are even fewer reports that thoroughly compare 2D and 3D cultures. The limited number of studies is probably also due to the high technical demands of 3D culture protocols. For example, Hoefner et al. [35] demonstrated a dynamic shift in the composition of the extracellular matrix (ECM) during adipogenic differentiation in 3D spheroids formed by a fluid overlay technique that did not occur in 2D cultivation. The ECM composition obtained in 3D cultivation resembled that of native adipose tissue. In this setting, the 3D spheroids required a significantly shorter adipogenic stimulus to maintain adipogenesis than in the 2D cultures. Kim et al. [36] demonstrated that 3D spheroids of MSC from dental follicles, generated with a newly developed microchip dish, showed a higher level of pluripotency markers compared with those in a 2D culture, whereas the degree of adipogenic differentiation in these cells remained unaffected by the cultivation dimensionality.

From the studies cited above, it is apparent that limited information is available on adipogenic differentiation of adMSC in 3D cultures. In particular, to our knowledge, the release of adipokines and other relevant differentiation factors has not yet been investigated. With the need for a robust and reproducible 3D cultivation method and in order to obtain 3D spheroids whose size can be precisely adjusted at the beginning of the experiment by inserting an exact number of cells, we applied a relatively novel method of spheroid generation based on a magnetic bead technique [37,38,39]. Using this cell culture model, we gained detailed insight into the effects of the 3D cell arrangement of adMSC upon adipogenic stimulation and on their adipokine release, as well as on inflammatory response during the differentiation process. Our results may also provide further indications of how these cells might behave in a 3D tissue environment during future therapeutic applications.

## 2. Materials and Methods

### 2.1. Tissue Donors

All donors gave written, informed consent to donate tissue to perform this study. The ethics vote was positively approved by the ethics committee at the Medical Faculty of the University of Rostock in compliance with the applicable law and is filed under the registration number A2013-0112 and A2019-0107. In total, material from 20 donors was used. The median age of the donors was 44 years (ranging from 30 to 67 years).

### 2.2. adMSC Isolation and Cell Culture

The adipose tissue was obtained by liposuction and the aspirated tissue was transported to the processing site at room temperature (RT) by direct transport or overnight and processed after 3 to 24 h. The processing and isolation of adMSC was performed according to the protocol of Meyer et al. [40] that was previously established by our group. Briefly, adipose tissue was cleared of tumescent solution by aspiration and 30 mL of adipose tissue was digested, shaking at 37 °C with 0.15 U/mL collagenase (NB4, Serva Electrophoresis GmbH, Heidelberg, Germany). The tissue was then filtered through a 100 µm cell strainer (Greiner Bio-One GmbH, Frickenhausen, Germany). The filtered tissue was washed with 10 mL Dulbecco’s phosphate buffered saline (DPBS, PAN Biotech, Aidenbach, Germany) + 10% fetal calf serum (FCS, PAN Biotech, Aidenbach, Germany) and sedimented for an additional 10 min. The aqueous infranatant containing the stromal vascular fraction (SVF) was filtered through a 40 µm cell strainer (Greiner Bio-One GmbH, Frickenhausen, Germany). Both the supernatant containing the lipid phase and the aqueous subsurface were centrifuged at 400× *g* for 10 min at room temperature. The supernatant was discarded and the two cell pellets were resuspended in a total of 10 mL PBS + 10% FCS and pooled. The cell suspension was centrifuged again for 5 min at 400× *g* and RT, and the supernatant was discarded. The pellet was resuspended in 12 mL of Dulbecco’s Modified Eagle Medium (DMEM, Thermo Fisher Scientific, Schwerte, Germany) + 10% FCS + 1% penicillin/streptomycin (penicillin: 100 U/mL; streptomycin: 100 mg/mL, Thermo Fisher Scientific, Schwerte, Germany, hereinafter referred to as complete culture medium). Cultivation took place in 75 cm^2^ TCPS flasks (Greiner Bio-One GmbH, Frickenhausen, Germany)) at 37 °C and 5% CO_2_. After 24 h of cultivation, CD34 positive cells were isolated using magnetic beads (CD34 positive isolation kit, Thermo Fisher Scientific, Schwerte, Germany). The number of beads was based on the confluence of the cells and ranged from 10–25 µL of the bead suspension. The resulting cell population is the adipose-tissue-derived MSC (adMSC). For 2D culturing, 6800 cells were seeded per well (corresponds to 20,000 cells/cm^2^, 96 well, Greiner Bio-One GmbH, Frickenhausen, Germany). The 2D culture was cultivated up to 28 days with a medium change every 2–3 days. Cultivation was performed without specific stimulation (unstimulated) and with adipogenic stimulation (described in more detail in Section 2.4).

### 2.3. Three-Dimensional Cell Culture

The 3D cell constructs (hereinafter referred to as spheroids) of adMSC were generated by using magnetizable nanoparticles (n3D, Bioscience, Greiner Bio-One GmbH, Frickenhausen, Germany; nanoparticle diameter 50 nm, composed of gold, poly-L-lysine, and iron oxide). For this purpose, adMSC in a 75 cm^2^ cell culture flask growing with 80–90% confluency were incubated overnight with 80 µL of the magnetizable nanoparticles in complete culture medium. This incubation causes the nanoparticles to attach electrostatically to the cell membrane (verified by phase contrast microscopy). The nanoparticle-equipped cells in the TCPS flask were washed twice with PBS and then incubated with trypsin containing ethylenediaminetetraacetic acid/EDTA (Gibco/Thermo Fisher Scientific, Schwerte, Germany) for 5 min at 37 °C. The detached cells were suspended in complete culture medium and centrifuged at 400× *g* for 5 min. The supernatant was discarded, and the cell pellet resuspended in 5 mL complete culture medium. The resulting cell suspensions were stored on ice until measurement. The cell suspensions were analyzed for the number of viable and dead cells with volume-calibrated cassettes (Via1-Cassette^TM^) and the NucleoCounter^®^ NC-3000^TM^ (both Chemometec, Allerod, Denmark). This procedure was performed according to the manufacturer’s protocol. For development of the 3D spheroids, 100,000 cells per well (96-well, Greiner Bio-One GmbH, Frickenhausen, Germany) were seeded. The nanoparticle-equipped cells were subsequently exposed in cell repellent plates to a magnetic field below the plate and spheroid formation was achieved overnight. The spheroids were cultivated for up to 28 days without a magnetic field. Every 2–3 days, the spheroids were exposed to the magnetic field in order to change the medium.

### 2.4. Adipogenic Stimulation

The procedure of adipogenic stimulation was described earlier [40,41]. In brief: after having prepared the confluent monolayers for 2D cultures and the 3D spheroids, adipogenic differentiation was induced after 3 days by adding a differentiation-stimulating medium, i.e., a complete culture medium containing 1 µM dexamethasone, 500 µM IBMX (3-isobutyl-1-methylxanthine), 200 µM indomethacin, and 10 µM insulin (all from Sigma-Aldrich Chemie GmbH, Taufkirchen, Germany). Adipogenic stimulation (AS) took place with every replacement of the medium three times a week (every second or third day). The replacement of medium for both differentiating adMSC and undifferentiating adMSC without specific differentiation factors (unstimulated/US) was performed simultaneously.

### 2.5. Determination of Cell Numbers and Cell Analysis

To determine and analyze the cell numbers of adMSC after different cultivation approaches, the NucleoCounter^®^ NC-3000^TM^ (Chemometec, Allerod, Denmark) aggregated cells assay was used. Cells were incubated with trypsin/EDTA (Gibco) at 37 °C (2D culture: 5 min; 3D culture: 20 min), dissociated, and the detachment reaction was stopped with the same volume of complete culture medium. The resulting cell suspensions were stored on ice until measurement was performed. The measurement of the cell numbers and cell diameters was performed according to the manufacturer´s protocol. The respective cell suspension was analyzed for the number of viable and dead cells with volume-calibrated cassettes (Via1-Cassette^TM^) and the NucleoCounter^®^ NC-3000^TM^ (both Chemometec, Allerod, Denmark) according to the manufacturer’s protocol (*n* = 9).

### 2.6. Analysis of 3D Culture Spheroids

Spheroids were visualized using phase contrast microscopy in an appropriate culture medium (Carl Zeiss Microscopy Deutschland GmbH, Oberkochen, Germany; Axiovert 25, 10x objective, *n* = 3) after 1, 14, and 28 days under unstimulated cultivation conditions and under adipogenic stimulation. For the determination of the size of the spheroids, the phase contrast images (Zeiss Zen Blue) were converted to binary data using ImageJ and calculated in terms of the area of the largest diameter of the respective spheroid. Zeiss Zen lite software was used to determine the spheroid diameter.

### 2.7. Preparation of Microscopic Slides for Histological Analysis

Three-dimensional cultured spheroids were washed with PBS after 14 and 28 days of adipogenic stimulation (AS) or unstimulated cultivation conditions (US) and fixed in 4% paraformaldehyde (PFA) at 4 °C overnight. After washing again, samples were dehydrated in 30%, 50%, 70% and absolute ethanol for 30–60 min each. Until embedding them in LR White acrylate resin (medium grade, Plano GmbH, Wetzlar, Germany), the samples were stored in absolute ethanol. For embedding, the specimens were infiltrated with a 1/1 mixture of ethanol/LR White in an open vial after dehydration overnight. Subsequently, infiltration with pure LR White was performed for 4 h. Samples were transferred to gelatin capsules, filled with LR White, hermetically sealed, and polymerized at 50 °C for approximately 2 days. After embedding in LR White resin blocks, thin sections with a section thickness of 0.5 µm were prepared with an ultramicrotome (Ultracut S, Reichert/Leica, Vienna) using a diamond knife (Diatome, Nidau, Switzerland). The sections were stained with toluidine blue and covered in mounting medium for analysis with a light microscope (Zeiss Axioskop 40, *n* = 3) equipped with a digital camera (Zeiss Axiocam ERc 5s, both Carl Zeiss Microscopy Deutschland GmbH, Oberkochen, Germany).

### 2.8. Immunofluorescence Staining and Image Analysis

After 14 and 28 days under corresponding stimulation, the samples were stained with calcein acetoxymethyl ester (Calcein AM, AAT Bioquest, Sunnyvale, CA, USA), propidium iodide (PI, Thermo Fisher Scientific, Schwerte, Germany), and Bis-benzimide H33342 trihydrochloride (Hoechst 33342, Sigma-Aldrich Chemie GmbH, Taufkirchen, Germany) to determine their viability. For this purpose, cells were washed with PBS and then incubated in complete culture medium with calcein AM 1:3000, PI 1:50, and Hoechst 33342 1:2000 for 20 min at 37 °C. After incubation, samples were washed and visualized in PBS (PAN Biotech, Aidenbach, Germany) on the inverted confocal laser scanning microscope LSM 780 (Carl Zeiss). Images were taken and analyzed using ZEN 2011 software (black edition, both Carl Zeiss Microscopy Deutschland GmbH, Oberkochen, Germany). For the spheroids from the 3D culture, several images were taken with the Z-stack function (slices with an interval of 5 µm, pinhole 1 AU) and a 3D reconstruction was made by an overlay. The overlay was used to illustrate the distribution over the entire spheroid (*n* = 3).

After 14 days of adipogenic stimulation or unstimulated cultivation conditions, samples were stained with Bodipy (Thermo Fisher Scientific, Schwerte, Germany) and Hoechst 33342 to detect fat vacuoles and for localization. For this purpose, the samples were washed with PBS and dist. H_2_O at the time of measurement, then fixed with 4% PFA for 1 h, washed again, and incubated with Bodipy 1:250/ Hoechst 33342 1:2000 in complete culture medium at 37 °C for 20 min. After washing again with H_2_O and PBS, the samples were visualized on the inverted confocal laser scanning microscope LSM 780 as described above (*n* = 3). 

### 2.9. Quantification of Adipokines by Multiplex Analysis

The expression of a set of adipokines: adipogenesis-related factors, nerve growth factor (NGF), interleukin 6 (IL-6), leptin, IL-8, hepatocyte growth factor (HGF), adiponectin, monocyte chemotactic protein 1 (MCP-1), tumor necrosis factor (TNF), resistin, IL-1β, and plasminogen activator inhibitor 1 (PAI-1, total) was quantified in both supernatants and lysates of 2D and 3D cultured adMSC using the MILLIPLEX^®^ MAP Kit for Human Adipocyte Magnetic Bead Panel (Merck Millipore/Sigma-Aldrich Chemie GmbH, Taufkirchen, Germany) according to the manufacturer’s instructions. The samples were collected after 7 and 14 days of unstimulated and adipogenically stimulated cultures, respectively. Lysates were prepared using the Bio-Plex^®^ Cell Lysis Kit (Bio-Rad Laboratories GmbH, Feldkirchen, Germany). Briefly: standards, controls, and samples were incubated together with the antibody bead mix in a 96-well plate at 4 °C for 16–18 h. The plate was then incubated with detection antibody at RT for 1 h, followed by incubation at RT with streptavidin-phycoerythrin for 30 min. Fluorescence intensity minus background intensity and the resulting protein concentrations of biomarkers were measured using the Bio-Plex^®^ 200 System and Bio-Plex^TM^ Manager 4.1.1 software (both Bio-Rad Laboratories GmbH, Feldkirchen, Germany). Protein concentrations were normalized to the respective cell number, which was measured using NC-200 (*n* = 4).

### 2.10. Statistical Analysis

Sample size *n* included at least 3 and a maximum of 9 biological replicates. The respective n number is given in the associated figure. Depending on the assay and the associated accuracy, each sample was measured in double or triple technical replicates. Microsoft Excel 2010 and GraphPad Prism 7 software were used for statistical analysis. Data were presented as median values displayed as boxplots. One-way ANOVA post hoc uncorrected Fisher´s LSD was used for analysis of the spheroid area, analysis of the spheroid diameter, and protein expression and Friedman Two-way ANOVA post hoc test for cell number and cell diameter. The probability value of *p* < 0.05 was set as the significant difference (indicated by §, #, and * in the graphs). All graphs were prepared using GraphPad Prism 7 software (GraphPad Software, San Diego, CA, USA).

## 3. Results

### 3.1. Phenotype of 3D Cultured Unstimulated and Adipogenically Stimulated adMSC 

The spheroids were visualized under unstimulated cultivation conditions and under adipogenic stimulation over a period of 28 days by phase contrast microscopy (Figure 1a). After only 24 h, uniformly sized, symmetrically shaped, compact spheroids were generated with the help of this new 3D cultivation method. In the following, under non-stimulated (control) cultivation conditions, spheroids became smaller over the 28-day culture period (Figure 1a, left), whereas the size of adipogenically stimulated spheroids appeared stable within the observed timeframe (Figure 1a, right). Measurements of the spheroid area and diameter by image analysis confirmed our observations that the unstimulated spheroids became smaller over the observation period, whereas the size of the adipogenically stimulated spheroids remained largely unaltered (spheroid area: Figure 1b and spheroid diameter: Figure 1c).

To obtain information on the viability and distribution of adMSC in 3D, the spheroids were fluorescently labeled by live/dead staining after 14 and 28 days under corresponding stimulation (Figure 1d and Appendix A). In both unstimulated and adipogenically stimulated spheroids, the nuclei (blue) were arranged in a compact manner and evenly distributed. The majority of cells were alive (green), but dead cells (red) were also detectable and evenly distributed. Consistent with the results of phase contrast microscopy, vital staining showed that the size of the adipogenically stimulated spheroids remained largely stable during the 28-day experimental period, while the unstimulated spheroids became smaller and appeared less intact at the end of the experimental period. However, there were no obvious changes in the proportion of live and dead fractions and their spatial distribution (Appendix A). Due to the penetration depth of the laser and the compactness of the spheroid, only the outer shell of the spheroid could be depicted. The core of the complete spheroid could not be visualized by confocal microscopy and was therefore examined in detail by histology (see Section 3.4).

### 3.2. Dependence of the Cell Number Development and the Single Cell Diameter on the Cell Culture Model and Differentiation

The development of the cell number in the course of cultivation provides information about the proliferation behavior of the cells. We were able to show that a large portion of the cells in 3D cultivation perished in the first week of cultivation: with an initial seeding of 100,000 cells/well, only about 25% (US) and 40% (AS) could be detected after 7 days. Furthermore, the progression of the cell number under unstimulated cultivation conditions in 3D culture was clearly different from the progression in 2D (Figure 2a): while the cell quantity in 2D increased steadily over the observation period of 28 days (from approx. 14,000 cells/well on day 7 by about 2.5 times to approx. 34,000 cells/well on day 28), there was a steady decrease in the cell quantity in 3D (a 3-fold decrease from approx. 25,000 cells/spheroid to approx. 8500 cells/spheroid).

In the adipogenic differentiation, the cell number development was different from that in the unstimulated condition. Whereas in the 2D cultivation the cell numbers almost stagnated over the observation period, in the 3D cultivation there was an apparent decrease in cell numbers from an average of about 42,000 cells/spheroid to approx. 6500 cells/spheroid, representing a 6-fold decrease, which was significant on day 28 of the experiment (Figure 2a). 

Interestingly, the single cell diameters differed depending on cultivation conditions. While in the 2D cultivation both the unstimulated and the adipogenically stimulated cells had a similar cell diameter of approx. 17–19 µm (median), the cell diameter in the 3D cultivation was significantly smaller, reaching approx. 12–14 µm (Figure 2b).

Because we observed a significant decrease in cell numbers in the 3D cultures after longer cultivation periods, we focused further investigations on the earlier to the intermediate time point of differentiation after 7 and 14 days of culture. However, for comparison, we additionally examined aspects of spheroid development after a longer 3D culture of up to 28 days.

### 3.3. Adipogenic Differentiation of adMSC in 2D and 3D Cultures

Adipogenic differentiation was demonstrated by fluorescence imaging of lipid accumulation in adMSC under both 2D and 3D cultivation using Bodipy lipid stain. Knowing the limited penetration of the laser, we analyzed the spheroids in toto as a first step. After 14 days of cultivation, the unstimulated cells showed no clear lipid accumulation in either the 2D or the 3D culture (Figure 3). In contrast, adipogenic stimulation led to the formation of relatively evenly distributed lipid accumulations (yellow) in both the 2D and 3D culture (Figure 3a,b). As already indicated by the live/dead staining (Figure 1a), a densely packed cell arrangement in the spheroid shell independent of the differentiation status is visible (Figure 3b). In contrast, under 2D culture conditions the cells are spread at a lower density (Figure 3a). 

### 3.4. Morphology and Adipogenic Differentiation of the Spheroids Shown by Histological Sections

In order to study the structure, degree of differentiation, and distribution of lipid accumulations in more detail on the cellular level in all areas of the spheroids, we further evaluated sections of resin-embedded spheroids. The resulting sections with a thickness of 0.5 µm were stained with toluidine blue, which provided an excellent resolution in the Z-Axis and enabled an overview of the structures as well as a high resolution of the lipid inclusions. 

Microscopic analysis of the sectioned spheroids showed a regular outer surface lining after 14 days of treatment (Figure 4a,b). Using higher magnification with oil immersion, it became apparent that both the unstimulated and adipogenically stimulated spheroids had formed two zones with different cellular arrangements (Figure 4c,d). With both treatments, an outer, enveloping layer characterized by cells with a flat phenotype was observed (Figure 4c,d; marked with arrows). The outer cellular envelope tended to be multilayered. In contrast, the cells inside the spheroids generally appeared more round-shaped and there was a clear difference in the appearance of the unstimulated compared with the adipogenically stimulated 3D culture conditions (Figure 4e,f). Whereas the interior of the unstimulated spheroids presented with relatively few intact cells and occasionally with cell-free areas containing cellular debris (Figure 4c,e; cell debris marked with asterisks), the interior of the adipogenically stimulated spheroids was frequently filled with large cells that exhibited a considerable number of vacuoles of varying sizes (Figure 4d,f; vacuoles marked with arrowhead). These vacuoles appeared as bubble-like, foamy inclusions within the cells, strongly suggesting an accumulation of lipid droplets, indicating differentiation of these cells into (pre-)adipocytes. In contrast, only a few rather small inclusions are visible in the unstimulated spheroids at the same magnification. Thus, adipogenically differentiating spheroids appeared structurally more homogeneous with clearly demarcated cellular territories and without evidence of cell death, whereas in the unstimulated spheroids, cellular debris could already be observed relatively close to the shell region. However, a necrotic core was not apparent in the unstimulated spheroids, as intact cells could be detected up to the center of the spheroid. Moreover, even after 28 days of stimulation, at a time when unstimulated spheroids displayed an even greater degeneration, no necrotic core was found in either unstimulated or adipogenically stimulated spheroids. In general, the zoning of the spheroids with an outer shell and core region, which was evident after 14 days of stimulation, is largely preserved during longer cultivation (Appendix A, spheroids after 28 days). Cellular debris already visible in the unstimulated spheroids after 14 days was even more pronounced after 28 days of culture and the remaining cells appeared loosely arranged. In contrast, less debris was observed in the adipogenically stimulated spheroids, which appeared more compact and remained structurally intact.

The nanoparticles used to develop the spheroids were visible in both stimulation conditions (brown coloring) and, after 14 days, they tended to form small clusters that may also reside intracellularly but were predominantly visible in the shell region (Figure 4). After 28 days of stimulation, the nanoparticles tended to form larger, meandering accumulations that mainly covered the shell and the subjacent region (Appendix A).

### 3.5. Changes in the Release of Adipokines Dependent on 2D and 3D Cultivation and the Differentiation Status

The expression of a set of adipokines was measured in the supernatants and lysates of 2D and 3D cultured adMSC after 14 days of cultivation by a multiplex assay. Relevant adipokines such as adiponectin, plasminogen activator inhibitor (PAI-1), interleukin 6 (IL-6), monocyte chemotactic protein 1 (MCP-1), hepatocyte growth factor (HGF), and leptin were differentially expressed dependent on both 2D and 3D cultivation and adipogenic differentiation (Figure 5 and Table 1). The amount of all adipokines was always considerably higher in the supernatants than in the respective lysates (Appendix A), and because there were no conflicting results between the amount of adipokine in supernatant and lysate, we focused on the supernatant in the presentation. For example, adiponectin was released exclusively during adipogenic stimulation of both 2D and 3D cultured adMSC. The adiponectin release was significantly higher in the 3D model than in the 2D model and increased with the progression of the adipogenic stimulation period (Figure 5a). 

The other adipokines detected were primarily present in larger amounts under 2D culture conditions. For example, PAI-1 release is much higher under 2D conditions and can only be increased slightly by adipogenic stimulation (Figure 5b). In addition, the release decreased with the duration of cultivation. The release of the pro-inflammatory cytokines IL-6, IL-8, and MCP-1 was also significantly higher under 2D culture conditions (Figure 5c,d and Table 1). Unlike adiponectin, there was a decrease in these three cytokines over the period of 14 days of cultivation and also due to adipogenic stimulation. At a much lower release level, this development was also displayed under 3D cultivation conditions. HGF was released under 2D culture conditions in the unstimulated cultures in a manner dependent on the cultivation time (significant increase after 14 days compared with 7 days) (Figure 5e). The adipogenic stimulation led to a slight but non-significant increase in HGF release in the 2D culture. Under 3D culture conditions, there was a tendency towards reduction in release over time. Interestingly, leptin was detected almost exclusively under 2D culture conditions: this showed a higher level in the adipogenically stimulated cells (Figure 5f). In the 3D culture, only very low levels were detected after adipogenic stimulation.

Resisitin and the nerve growth factor NGF were detectable only at very low levels (Table 1). Here, the resistin release was increased by adipogenic stimulation in 2D. In contrast, the adipogenic stimulation did not affect the resistin release in the 3D culture. The NGF release was reduced by 3D cultivation and not affected by adipogenic stimulation. However, no statistical significance was found for either (Appendix A). In addition, only very low levels of interleukin-1β (IL-1β) and tumor necrosis factor (TNF) could be detected. In this case, this could not be influenced by adipogenic stimulation or by the dimension of cultivation (Table 1).

## 4. Discussion

The cultivation conditions are crucial for cell behavior in terms of proliferation, differentiation, and other specific cellular reactions such as the release of biologically active factors, synthesis of the extracellular matrix, or cell–cell interactions. The cultivation conditions are determined by the addition of specific soluble components (e.g., growth factors, hormones, and pharmacological compounds), insoluble substrates (e.g., artificial, semi-artificial and natural coatings), and the dimensionality of the culture (2D and 3D culture) [42]. The final outcome of the complex interplay of the aforementioned factors further depends on the concentration, the particular time of stimulation, and specific combination of factors, so prediction is often elusive. Specifically, knowledge of the effects of 3D cultivation on adipogenic differentiation is currently limited and constantly subject to new findings [17,43,44,45]. Therefore, in this study, we compared the effects of 2D and 3D cultivation in human adMSC without specific stimulation and under adipogenic stimulation, respectively, with focusing on differentiation and adipokine release. In the 3D model system used in this study, spheroid formation of adMSC was induced by cell-bound magnetic nanoparticles temporarily exposed to a magnetic field. The application of this technique led to the formation of spheroids with a uniformly defined size and compaction from the beginning of the experiment. In addition, applying the magnetic field during medium exchange prevents the loss of the spheroid due to aspiration. Different studies have shown that the nanoparticles used are biocompatible and have no effect on cell development in terms of viability, proliferation, or metabolism [46,47,48]. Similarly, we also found no adverse effects during cultivation (not even in the areas where an accumulation of nanoparticles occurred). Our observations confirmed these reports and so we conducted our experiments for up 28 days. During this period, the viability of cells in 2D culture and 3D spheroids was relatively high and a series of specific changes occurred depending on the stimulation and culture dimensionality.

### 4.1. Morphology of Unstimulated and Adipogenically Stimulated MSC Spheroids

This 3D spheroid model exhibited a clear difference in spheroid size depending on the differentiation protocol applied. While the unstimulated spheroids became smaller over the experimental period (from a diameter of 500 µm at the beginning to below 400 µm after 28 days), the size of the adipogenically stimulated spheroids remained relatively constant (at about 600 µm in diameter). In addition, the unstimulated spheroids showed evidence of disintegration (isolated cell debris and more loosely attached cells inside the spheroid), whereas the adipogenically stimulated spheroids remained structurally intact. These differences became more and more evident over the observation period of 28 days. Since the morphological differences between the undifferentiated and adipogenically differentiated spheroids extended to the center of the spheroid, we assume that the differentiation compounds (i.e., dexamethasone, IBMX, indomethacin, and insulin) penetrate the complete spheroids.

In spheroids of tumor cells, there is often a marked increase in the size of the spheroids when proliferation is strong. This increase in size is then associated with the formation of a hypoxic region, which ultimately then leads to the formation of an avital, necrotic core in the spheroids [49]. In our experiments, there was no increase in size over the experimental period under any of the treatments performed and a necrotic core did not develop. Moreover, despite their smaller size, cellular debris was mainly detectable in the unstimulated spheroids, whereas the adipogenically stimulated spheroids, which were even larger, had vital, well-differentiated cells throughout their nuclei.

Information on cell viability in spheroids from MSC is sparse and rather suggests the opposite behavior compared with spheroids from tumor cells. For example, Regmi et al. [50] described higher cell viability of MSC in a 3D culture model than with 2D culturing. In the process they observed that heme oxygenase 1 (HMOX1) was significantly upregulated in 3D cultivated MSC, indicating a hypoxic situation with an increased level of reactive oxygen species (ROS). By inhibiting the hypoxia cell signaling via the hypoxia-inducible factor-1 subunit alpha (HIF1α), a suppression of HMOX1 was caused. Similarly, Zhang et al. [51] also demonstrated an involvement of HIF1α in the MSC survival within 3D cultures, as the increase in the HIF1α expression was associated with increased resistance to apoptosis triggered by oxidative stress. These studies show that the viability of cells in spheroids is influenced by several factors, such as the diffusion of nutrients, oxygen, and degradation products through the spheroidal structure. Thus, the size of the spheroids is also decisive for the survival and differentiation of cells within the spheroid [52]. Specifically, in a recently published study by Schmitz et al. using human adMSC (the cell type used in our study), it was shown that not only the spheroid size but also the spheroid cultivation method plays a crucial role in the occurrence of critical hypoxia [53]. There were large differences in critical spheroid size depending on the spheroid culture method (from 180 µm on microstructured plates to 920 µm spheroids in the hanging drop technique). The spheroids presented in our work were, at most, just over 600 µm in size, so a critical undersupply and thus cell death in the core of the spheroids may not have occurred. In addition, the smaller spheroids (undifferentiated with 500 µm and smaller) show more indications of cellular degeneration and are more fragile than the larger spheroids (adipogenically differentiated at a size of about 600 µm). This suggests that the spheroid size plays a more minor role in the survival of adMSC in this type of 3D culture than in the differentiation state.

However, not only the number of cells but also the size of the cells can influence the size of the spheroid. The variation in single cell diameters in the different cultivation methods we observed was significant. The diameter of the cells in the 3D cultures was about 30% smaller than in the 2D culture. This is consistent with a number of studies, where a decreased cell volume of up to 75% was observed in 3D cultivation [52,54]. This reduction in cell size in the 3D culture is induced by cytoskeletal changes; Ruiz and Chen described a pivotal role of actinomyosin and the myosin-generated tension in the process of 3D spheroid organization [55].

However, what are the reasons for the differences in stability between unstimulated and adipogenically stimulated spheroids? In our study, spheroid size does not seem to be crucial. There might, however, be a link to a change in cellular energy metabolism. As we have previously shown, the metabolic basis of adMSC shifts significantly toward oxidative phosphorylation and β-oxidation under adipogenic differentiation compared with undifferentiated adMSC. Thus, the metabolic capacity shifts toward lipid metabolism [56]. However, because these studies of energy metabolism were performed under 2D rather than 3D conditions, a limitation of the transferability remains.

In principle, the adipogenic differentiation performed well in both 2D and 3D cultures under adipogenic stimulation. Metachromatic toluidine blue staining of the thin sections of these spheroids showed a large number of vacuoles in the cells just below the enveloping cell layer of the adipogenically stimulated spheroids. Vacuoles of this type did not occur in the unstimulated spheroids. This suggests substantial lipid accumulation in the adipogenically stimulated spheroids, which is consistent with our lipid detection results.

Interestingly, some cells are filled with numerous smaller vacuoles, while others contain only two to three large vacuoles. This suggests that after 14 days of stimulation, adMSC have differentiated toward a pre-adipocytic cell fate throughout the spheroid core [57]. It has been described that the degree of adipogenic differentiation is enhanced by the cell–cell and cell–matrix contacts as a result of the 3D arrangement [58,59]. In general, analyses of adipocytes and their differentiating progenitor cells are challenging because they are hampered by the high lipid content of the cells [58,60]. For example, we suspect that our cell counts of adipogenically stimulated adMSC may even be too low because the cells need to be concentrated by centrifugation and because especially the large, lipid-laden cells are then lost for quantification.

### 4.2. Adipokine Release of adMSC in 2D and 3D Culture

In our study, both 2D and 3D cultivation resulted in adipogenic differentiation after adipogenic stimulation, whereas adipogenic differentiation was absent in the unstimulated condition. This was the basis for a detailed comparison of adipokine release. We demonstrated that the adipokine released in the highest amounts was the protein hormone adiponectin. However, adiponectin release could only be induced by adipogenic stimulation; without specific stimulation, adiponectin release was negligible. Adiponectin, primarily produced in adipose tissue, causes a wide range of physiological effects, including, e.g., the regulation of glucose levels as well as fatty acid breakdown [61].

Although adipogenic stimulation appeared necessary for the onset of adiponectin expression in our study, 3D cultivation induced a further approx. 3-fold increase in adiponectin release compared with the 2D cultures. In addition, the duration of the adipogenic stimulation and thus the progression of adipogenic differentiation led to a significant increase in the amount of adiponectin released. The adipogenic stimulation compound used included indomethacin. Indomethacin strongly promotes adipogenesis through a mechanism that increases two factors: the transcription factor CCAAT enhancer binding protein β and the nuclear receptor peroxisome proliferator-activated receptor γ2/PPAR γ2 [62]. Consequently, the further increase in the release of adiponectin in 3D cultures must be induced directly or indirectly by the dimensionality of the culture. A few aspects might be responsible for the increased adiponectin release in 3D. This may include the reduced cell size already mentioned [52] and the roundish cell shape in the 3D spheroid [63], which may also influence adipogenic differentiation.

Another adipokine analyzed was the protein hormone leptin. In our study, leptin could only be detected in relatively low amounts under 2D conditions. This showed increased release due to adipogenic stimulation on day 7. In the 3D culture, leptin release was negligible. Leptin plays an essential role in regulating fat metabolism in mammals and is expressed mainly by adipocytes. Leptin limits lipid storage by affecting specific metabolic pathways in adipose tissue. Leptin induces the release of glycerol from adipocytes and inhibits the synthesis of fatty acids [64]. Thus, the low leptin release in 3D cultures may further indicate the promotion of a cellular status for an unimpeded adipogenic differentiation process in spheroids.

In principle, MSC have been shown to possess anti-inflammatory and immunomodulatory properties. In our experiments, we showed that the release of the inflammation-related adipokines PAI-1, IL-6, IL-8, and MCP-1 was markedly reduced during 3D cultivation compared with 2D. This is consistent with the outcomes of other 3D/MSC cultivation studies. In this respect, Bartosh et al. [54] demonstrated that MSC in spheroids express high levels of stanniocalcin-1, a protein with anti-inflammatory and anti-apoptotic properties. Lee et al. [65] also showed an increase in anti-inflammatory factors through the spheroid formation of MSC, such as the IL-10.

Strikingly, the release of the pro-inflammatory cytokines/chemokines IL-6, IL-8, and MCP-1 was even lower when the cells were adipogenically stimulated (in both 2D and 3D cultures). This downregulation of IL-6 is in contrast to the results of the study conducted by Munir et al. [66], who showed a significantly higher IL-6 release in adipogenically differentiated MSC compared with the undifferentiated MSC. However, the cultivation and stimulation conditions were markedly different from those in our study. In addition, the literature on this aspect is sparse.

## 5. Conclusions

Our experiments have shown that the chosen 3D spheroid model is a reliable and valuable tool to study adMSC differentiation because it allows the generation of uniformly- and symmetrically-shaped spheroids that can be studied over a comparatively long culture period of 4 weeks. The findings based on these experiments provide new insights into the effects of cell arrangements of adMSC in 2D and 3D cultivation, on differentiation and pro-inflammatory activation. The dimensionality of cultivation revealed marked differences in the release of adiponectin and leptin, indicating an increased adipogenic differentiation capacity in 3D cultures. At the same time, it was again clearly shown that the release of pro-inflammatory factors was reduced by 3D cultivation. Currently, there is little knowledge of the underlying mechanisms causing the differences in the release of adipokines between 2D and 3D cultures. In addition, there has been only limited information on the signaling and control of MSC survival and proliferation in 3D cultures. Therefore, further efforts are needed to study the mechanisms and interactions in more detail. Such knowledge is not only important for the use of adMSC in cell therapeutic applications in a very complex 3D environment, but could also contribute to an understanding of the background of adipose tissue diseases.

## Figures and Tables

**Figure 1 cells-11-01313-f001:**
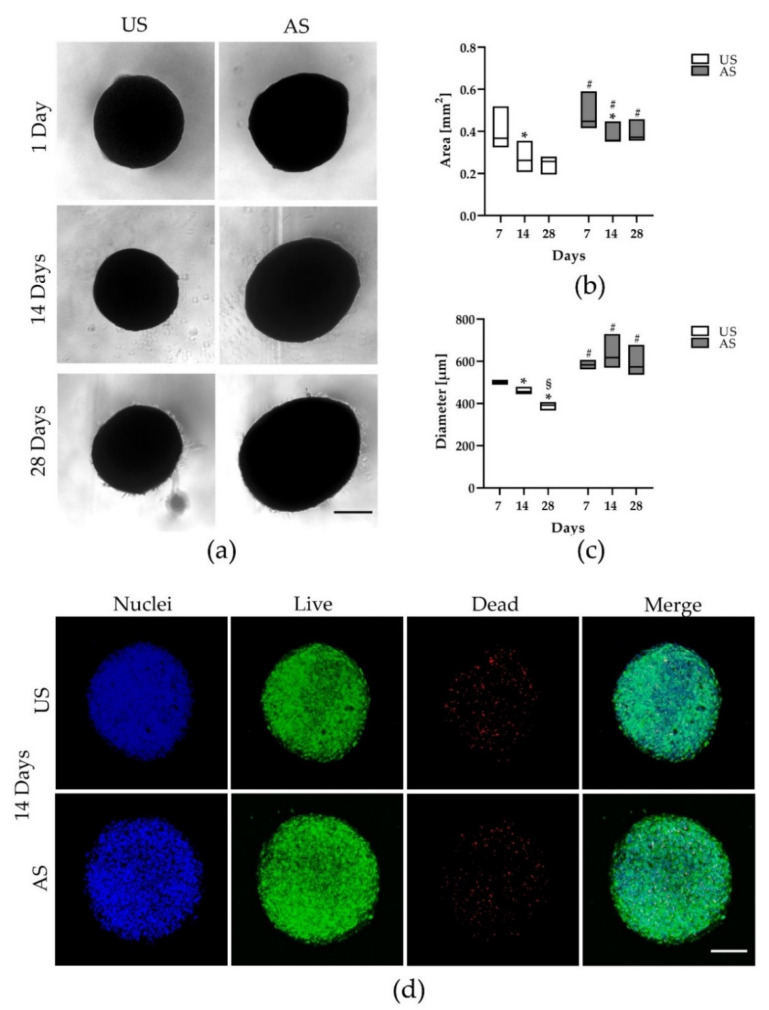
Analysis of spheroid morphology and size. (**a**) Imaging of unstimulated (US) and adipogenically stimulated (AS) spheroids after 1, 14, and 28 days (*n* = 3, phase contrast, Axiovert 25, scale bar: 200 μm). (**b**) Analysis of the spheroid area and (**c**) the spheroid diameter of unstimulated and adipogenically stimulated spheroids after 7, 14, and 28 days (* significantly different from day 7 of the respective stimulation, * *p* < 0.05; # significantly different from the respective time point of unstimulated culture, # *p* < 0.05; § significantly different from day 14 of the respective stimulation, § *p* < 0.05; One-way ANOVA post hoc uncorrected Fisher´s LSD with *n* = 3 for area measurement by image analysis using ImageJ and *n* = 4 for diameter measurement by image analysis using Zeiss Zen lite). (**d**) Live/dead staining of an unstimulated and an adipogenically stimulated spheroid after 14 days (*n* = 4, LSM 780, Zen black software, overlay of Z-stack, green: live, red: dead, blue: nuclei; scale bar: 200 μm).

**Figure 2 cells-11-01313-f002:**
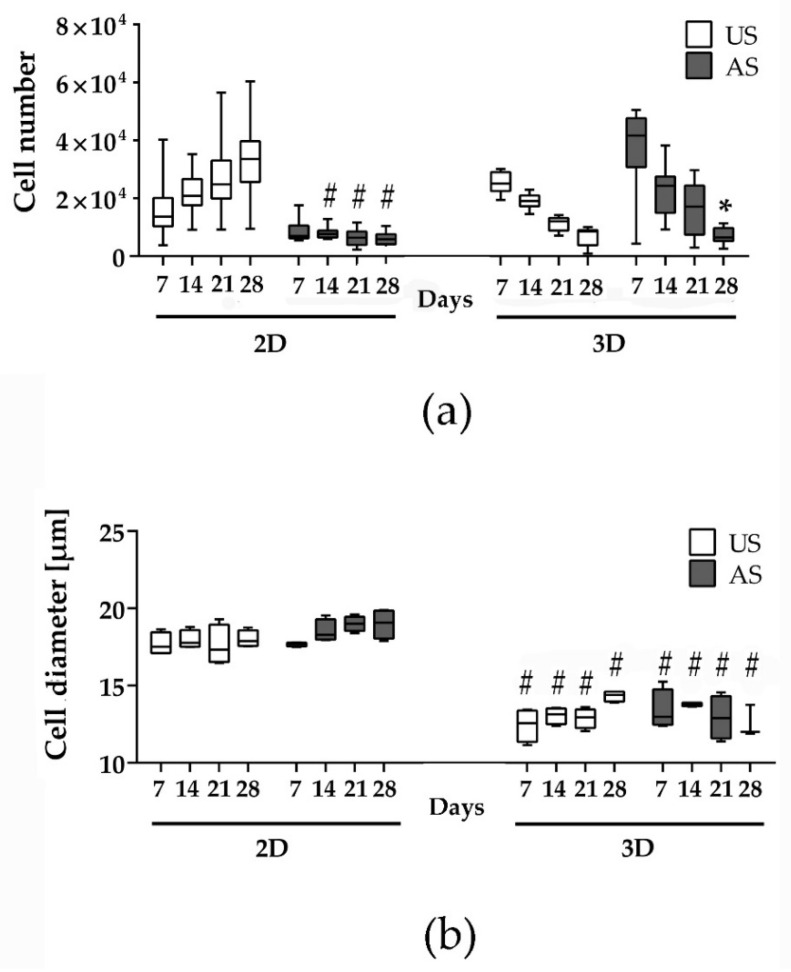
Quantification of the cell number and single cell diameter of adMSC in 2D and 3D cultures up to 28 days. (**a**) Cell number of unstimulated (US) and adipogenically stimulated (AS) adMSC within 28 days (*n* = 9; # significantly different to US in 2D culture at the respective time point, Two-way ANOVA post hoc uncorrected Fisher´s LSD; # *p* < 0.05; * significantly different to day 7 of adipogenically stimulated 3D culture, Friedman test post hoc Dunn´s multiple comparison test, * *p* < 0.05). (**b**) Diameter of single cells under US and AS conditions in 2D and 3D culture (*n* = 4; # significantly different from 2D culture at the respective time point, Two-way ANOVA post hoc uncorrected Fisher´s LSD; # *p* < 0.05).

**Figure 3 cells-11-01313-f003:**
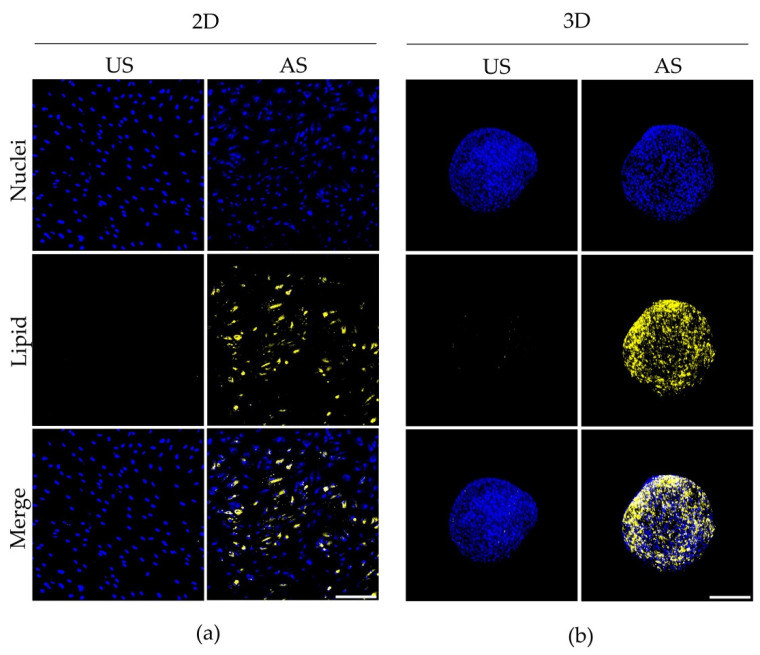
Lipid accumulation in unstimulated (US) and adipogenically stimulated (AS) adMSC after 14 days in (**a**) 2D and (**b**) 3D cultures (representative images from *n* = 3; lipid accumulation: yellow; nuclei: blue; confocal microscopy; scale bar: 200 μm).

**Figure 4 cells-11-01313-f004:**
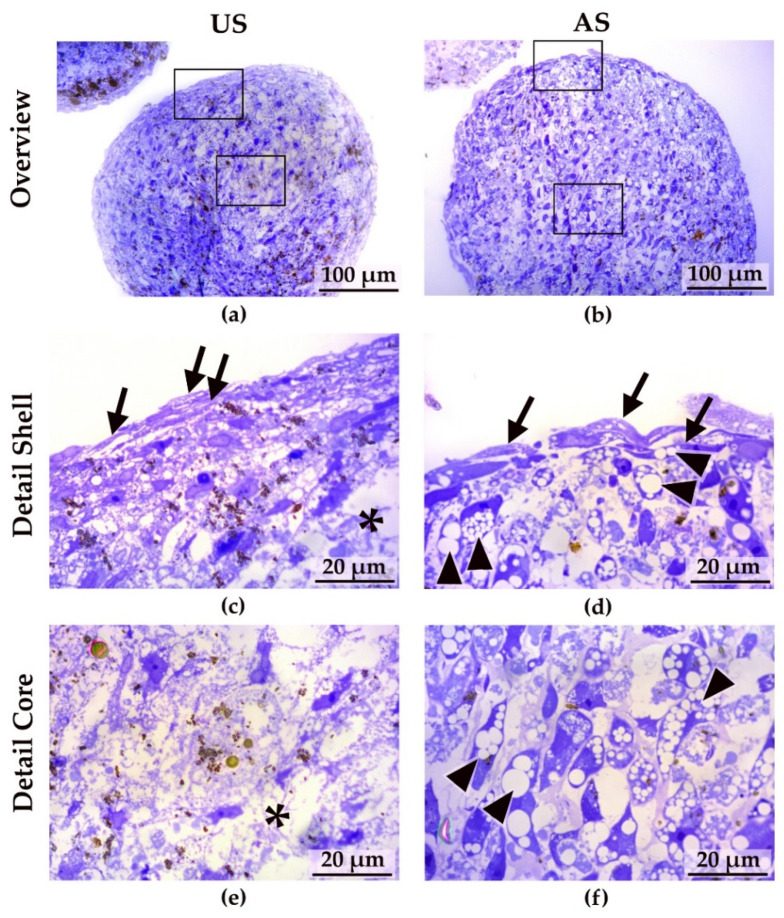
Thin sections (0.5 µm) of unstimulated and adipogenically stimulated 3D spheroids after 14 days. Overview images of (**a**) an unstimulated (US) spheroid and (**b**) an adipogenically (AS) stimulated spheroid (scale bars: 100 µm, frames in the overview images correspond to the areas shown below in high magnification, (**c**) detail of the shell region of an unstimulated spheroid (arrows: enveloping cells, asterisks: cell debris), (**d**) detail of the shell region of an adipogenically stimulated spheroid (arrows: enveloping cells, arrowheads, vacuoles), (**e**) detail of the core of an unstimulated spheroid (asterisks: cell debris), (**f**) detail of the core region of an adipogenically stimulated spheroid (arrowheads: different sized vacuoles; scale bars: 20 µm; representative images of 3 experiments; light microscopy (Zeiss Axioskop 40), toluidine blue staining).

**Figure 5 cells-11-01313-f005:**
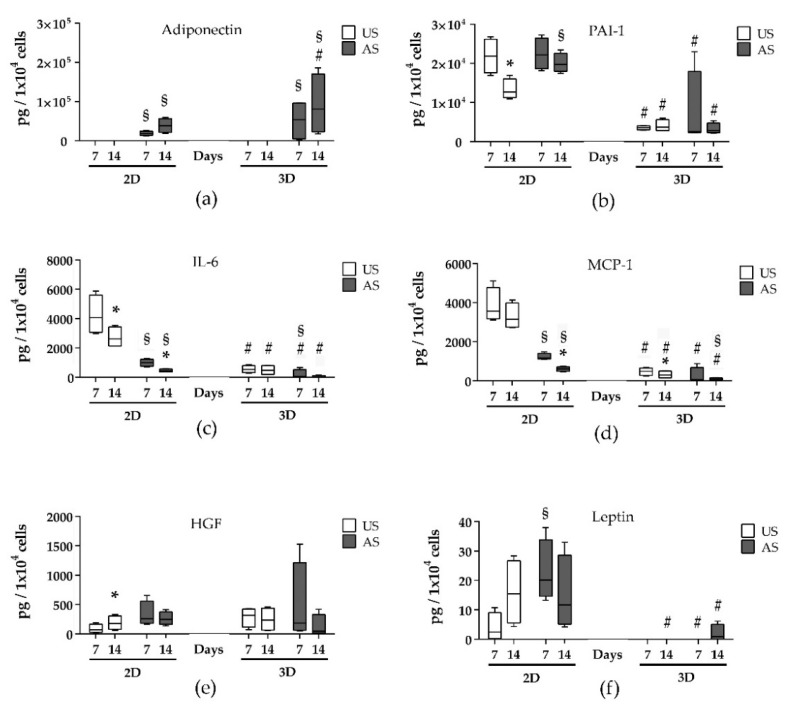
Quantification of adipokines in supernatants of unstimulated (US) and adipogenically (AS) stimulated 2D and 3D cultures after 7 and 14 days. (**a**) Adiponectin; (**b**) PAI-1; (**c**) IL-6; (**d**) MCP-1; (**e**) HGF; (**f**) leptin (concentrations were normalized to cell number, *n* = 4; # significantly different from 2D cultures at the respective time point; * significantly different from day 7 under the same cultivation conditions; § significantly different from US culture at the respective time point and dimension of cultivation; One-way ANOVA and Two-way ANOVA, post hoc uncorrected Fisher’s LSD; # *p* < 0.05; * *p* < 0.05; § *p* < 0.05).

**Table 1 cells-11-01313-t001:** Quantification of a subset of adipokines in cell culture supernatants of unstimulated (US) and adipogenically (AS) stimulated 2D and 3D cultures (multiplex assay, biological replicates: *n* = 4, technical replicates were involved in the calculations, median, values in pg/10^4^ cells, descending order depending on detected quantity, with decimal place in the single-digit range).

		2D		3D	
Analyte		US	AS	US	AS
Adiponectin	Day 7	0	17,923	0	53,454
	Day 14	0	38,056	0	150,746
PAI-1	Day 7	21,983	22,154	3809	2435
	Day 14	12,219	20,027	3702	2791
IL-6	Day 7	4010	1006	601	51
	Day 14	2561	423	485	36
MCP-1	Day 7	3565	1166	522	59
	Day 14	2988	598	305	73
IL-8	Day 7	276	86	81	22
	Day 14	418	71	40	20
HGF	Day 7	62	261	305	140
	Day 14	181	243	232	49
Leptin	Day 7	2.5	20	0	0
	Day 14	15	13	0	0.7
Resistin	Day 7	0.5	2.5	0.7	0.6
	Day 14	0.1	1.8	0.3	0
NGF	Day 7	1.4	1.0	0.1	0.1
	Day 14	1.0	0.8	0.1	0.1
IL-1β	Day 7	0.1	0.3	0.1	0
	Day 14	0.1	0.3	0.1	0.1
TNF	Day 7	0	0.2	0	0
	Day 14	0	0.1	0	0

## Data Availability

All data analyzed during this study are included in the published article.

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
