# Peer review of "A Comparative Study on the Adipogenic Differentiation of Mesenchymal Stem/Stromal Cells in 2D and 3D Culture"

_cells, 2022, doi:10.3390/cells11081313_

Round 1

Reviewer 1 Report

The authors demonstrated with a comparative study between 2D and 3D adipose tissue derived MSCs, that the cultivation are crucial for cell behaviour. The authors analysed the production and the release of several adipokines revealing, in particular, an increase of adiponectin and leptin and a reduction of proinflammatory factors in 3D cultures. This study is simple but adequately described and conducted.

Author Response

Dear reviewer,

thank you very much for the evaluation and your positive assessments.

With kind regards
Kirsten Peters

Reviewer 2 Report

The Authors have made some revisions to the manuscript based on comments by reviewers. 

Author Response

(The authors gave the same response as above.)

Reviewer 3 Report

This paper is interesting, but the information on 3D culture or novelty is lacking. The current content is not enough for the publication.

Taken together, major revisions should be made before re-submission. The paper would be re-considered only when all the comments were responded.

1.

The authors should introduce the 3D culture of several cells or tissue regions, such as bone, cancer, skin, and stem cells, of course. The authors should clarify the novelty of this study on adipogenic differentiation in the 3D system. The novelty is not understood completely, so the authors must add some sentences in the introduction. To reduce the authors’ burden, I suggest the sentences or references to be added for the revision.

Sentences

“There are some reports on the 3D culture system of bone [], cancer [], skin [], or stem cells []. The systems can perform a comparative study between 2D and 3D cultures combined with stromal cells. However, it is a little difficult to……… Therefore, in this study, we designed the ………….”

Recent references

Bone (Review and research)

Int. J. Mol. Sci. 201819(5), 1285;
Materials Science and Engineering: C 94, 703-712

Cancer (Review and research)

Cancers 202012(10), 2754

Tissue Eng. Part C Methods 201925, 711–720 https://doi.org/10.1089/ten.tec.2019.0189

Skin (Review and research)

 https://doi.org/10.3389/fbioe.2018.00154

doi.org/10.1038/s41514-020-0042-x

Stem cells (Review and research)

Development (2018) 145 (5): dev156166.

Int. J. Mol. Sci. 201819(4), 936

Stem Cell Res Ther 9, 50 (2018). 

Regenerative Therapy 18, 516-522, 2021.

2.

Because the size of spheroids is about 600 µm, I think hypoxia-derived apoptosis must be induced, especially cells in the center of spheroids. The authors should investigate the cell viability, Ki67 staining, or live/dead assay. If the authors cannot, the points should be discussed.

3.

At the stimulation to spheroids, I think the stimulation would treat the cells on the surface of spheroids more than cells present in the center of spheroids. Is there a difference in efficiency between the surface and center of spheroids? The authors should clarify.

4.

The authors should investigate the time-course expression of stemness markers.

5.

The abbreviation should be clarified the first time.  

MSC in abstract means mesenchymal stem cells, right? However, in the introduction, MSC indicates the mesenchymal stem cells and stromal cells (Line 48). Overall, it is a little difficult to understand the manuscript clearly.

The written English of the manuscript should be considerably improved, and therefore, a revision by a native speaker is highly recommended.

6.

The importance of stromal existence is not understandable.

Author Response

Dear reviewer, 
thank you very much for the evaluation report. We have carefully worked through the outstanding questions and discussion points. You will find the answers in the attached file. 
We hope that the revised manuscript meets with your approval.
With kind regards
Kirsten Peters

Round 2

Reviewer 3 Report

The introduction should be improved.

However, what is the strength of this study by comparing other research papers?

Although the reviewer mentioned in the first round, the authors should clarify the novelty of this study in the 3D system by discussing the 3D systems with each tissue site. The novelty is not understood completely. 

Author Response

See attached word file
